# Impacts of hurricanes and disease on *Diadema antillarum* in shallow water reef and mangrove locations in St John, USVI

**Renee D. Godard**[1]*, **C. Morgan Wilson**[1], **Caleb G. Amstutz**[2☯], **Natalie Badawy**[1☯], **Brittany Richardson**[1☯]

**1** Departments of Biology and Environmental Studies, Hollins University, Roanoke, Va, United States of America, **2** Community School, Roanoke, Va, United States of America

☯ These authors contributed equally to this work.

* rgodard@hollins.edu

**Data Availability Statement:** This original data is available (10.5281/zenodo.8395238).

**Funding:** Funding for this research was provided by the Paula Pimlott Brownlee and Janet Spear

## Abstract

The 1983–1984 mortality event of the long-spined sea urchin *Diadema antillarum* reduced their population by up to 99% and was accompanied by a phase shift from coral dominated to algal dominated reefs in the Caribbean. Modest rebounds of *D. antillarum* populations in the Caribbean have been noted, and here we document the impacts of two major hurricanes (2017, Irma and Maria) and the 2022 disease outbreak on populations of *D. antillarum* found by targeted surveys in the urchin zone at nine fringing reef and three mangrove sites on St. John, USVI. *D. antillarum* populations at the reef sites had declined by 66% five months after the hurricanes but showed significant recovery just one year later. The impact of recent disease on these populations was much more profound, with all reef populations exhibiting a significant decline (96.4% overall). Fifteen months after the disease was first noted, *D. antillarum* at reef sites exhibited a modest yet significant recovery (15% pre-disease density). *D. antillarum* populations in mangrove sites were impacted by the hurricanes but exhibited much higher density than reef sites after the disease outbreak, suggesting that at *D. antillarum* in some locations may be less vulnerable to disease.

## Introduction

Prior to the early 1980s, long-spined urchins, *Diadema antillarum*, were abundant macroalgae grazers in the Caribbean [1–3] with common densities ranging from of 3 up to 71 urchins m$^{-2}$ [4]. So numerous were these organisms in some areas, localized removal was once considered for the establishment of underwater marine trails [5]. This species consumes benthic algae that can compete with coral for space and plays a critical role in preventing algal overgrowth on reefs [6–9], a role that became obvious when the species collapsed in the early 1980s [3, 4, 10]. In early 1983, populations of *D. antillarum* adjacent to the Panama Canal first exhibited signs of disease, and within 13 months populations across the breadth of the Caribbean had experienced mass mortality (96–99%) by an unknown pathogen that likely spread along ocean currents and in the ballast water of boats [3, 11, 12]. This decimation of *D. antillarum* was

Professorships at Hollins University as well as support from Tom and Anna Lawson. The funders had no role in study design, data collection and analysis, decision to publish, or preparation of the manuscript.

**Competing interests:** The authors have declared that no competing interests exist.

associated with a phase shift in the Caribbean from coral-dominated reefs to those occupied primarily by macroalgae [3, 7, 13–16].

After the mass mortality event in the mid-1980s, rapid recovery of *D. antillarum* was anticipated as the phase shift to macroalgae released the species from food limitations and its reproductive potential (a female can produce millions of eggs per spawn) was impressive [13, 17]. However, *D. antillarum* populations have, overall, shown only modest recovery, with most populations remaining at least an order of magnitude lower than pre-1983 [2, 3, 9, 18–23]. Several non-mutually exclusive hypotheses have been proposed to explain the limited recovery of *D. antillarum* and include: suppressed recruitment resulting from low population density and their asynchronous spawning behavior [3, 24], increased competition from vertebrate reef herbivores [25], reduced populations of mutually beneficial heterospecific echinoids [3], increased vulnerability to disease due to decreased immune function [26], increased predation pressure [27], and the loss of structural complexity which reduces the availability of daytime refugia [28]. Though the Caribbean-wide recovery has been limited, several studies have reported a more modest recovery of *D. antillarum* populations associated with a localized return towards a more coral dominated community [9, 15, 29–32].

The reported impacts of hurricanes on *D. Antillarum* populations in the Caribbean have been variable. Hurricane Allen (cat 5, 1980) significantly reduced *D. antillarum* density on shallow reefs (5–8 m) in Jamaica but did not impact populations in deeper (10–20 m) water [33]. Similarly, Hurricane Irma (cat 4, 2017) caused a significant decline in already depleted *D. antillarum* populations in the Florida Keys [34]. However, the increased mortality associated with Hurricanes Hugo (cat 4, 1989) and Earl (cat 4, 2010), did not result in significant declines in density of *D. antillarum* in Lameshur Bay, St. John, U. S. Virgin Islands (USVI) [2]. And after Hurricane Dean (cat 5, 2007), density in a robust Southern Mexico *D. antillarum* population remained stable [29].

In late January 2022, another die-off of *D. antillarum* was noted, this time originating near a harbor in St. Thomas, USVI. Within four months, signs of disease similar to those of 1983 (a lack of tube feet control, slow spine reaction and loss, followed by epidermal necrosis) had occurred in populations throughout the Caribbean (1,300 km N to S and 2,500 km E to W) [35]. Current molecular techniques, combined with a veterinary pathology approach, led to the identification of a scuticociliate that resembled *Philaster apodigitiformis* as the causative agent of the 2022 disease outbreak [36]. While the full extent of the 2022 die-off is not yet known, a closely monitored *D. antillarum* population at Saba (Caribbean, Netherlands) exhibited a 99% mortality rate [35]. Similarly, Levitan and colleagues [32] reported that *D. antillarum* populations followed since 1983 in Greater and Little Lameshur Bays, St John, USVI, had exhibited a 98% decline.

Before the 1983 disease episode, this well-studied Lameshur Bay population of *D. antillarum* had averaged 14.39 m$^{-2}$ but collapsed to 0.08 m$^{-2}$ by 1984 [13]. While the population here did recover, the *Diadema* density never rose above 1.15 m$^{-2}$ [2]. Given the modest recovery in this bay in St. John, we became interested in comparing population patterns with other sites on the island. As such, in 2017 we surveyed *D. antillarum* populations at shallow-water reef sites on both the north and south sides of the island, as well as along the fringes of mangroves (*Rhizophora mangle*). This initial survey occurred nine months before the island was severely impacted by two category 5 hurricanes. On September 6, the western eyewall of Hurricane Irma tracked over St. John with estimated sustained wind speeds of 185 mph (161 kt), gusts over 220 mph (191 kt), and an atmospheric pressure of 916 mb [37–39]. Two weeks later, Hurricane Maria passed 60 miles south of St. John with an estimated wind speed of 155 mph (135kt) and an atmospheric pressure of 920 mb [38, 40]. We continued to follow patterns in *D. antillarum* density at these reef and mangrove sites through 2023 (Fig 1). Given that two major

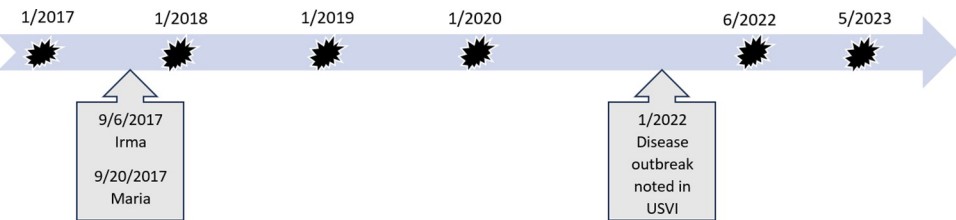

**Fig 1. Timeline.** Dates (month/yr) of *Diadema antillarum* surveys noted above the blue timeline, with dates of the two major hurricanes (month/day/yr) and the disease outbreak (month/yr) noted below the timeline.

perturbations occurred during that time interval, we were able to: 1.) compare the impacts of hurricanes and disease on *D. antillarum* populations; 2.) explore recovery patterns following these perturbations; and 3.) examine *D. antillarum* populations in multiple reef and mangrove locations.

## Methods

From 2017 to 2023 (except for 2021 due to COVID restrictions, Fig 1), we surveyed the abundance of *D. antillarum* at nine fringing reef sites (1–9) and three mangrove sites (M1-M3) in St. John, USVI (see Fig 2). The nine reef sites occurred within the boundaries of the Virgin Islands National Park and were originally chosen because of 1) their accessibility from land, 2) their coverage of both the north (sites 1–4) and south (sites 5–9) sides of the island, 3) inclusion of two areas surveyed on St. John since 1983 (site 5 is within the site "SQST" and 6 mirrors "DOBI" reported by Levitan [13]), and 4) their distance from one another (no two sites closer than 0.35 km). The three mangrove sites (Fig 2: M1-M3), separated by a minimum of 0.4 km, were in Hurricane Hole in the Virgin Islands Coral Reef National Monument and, prior to the

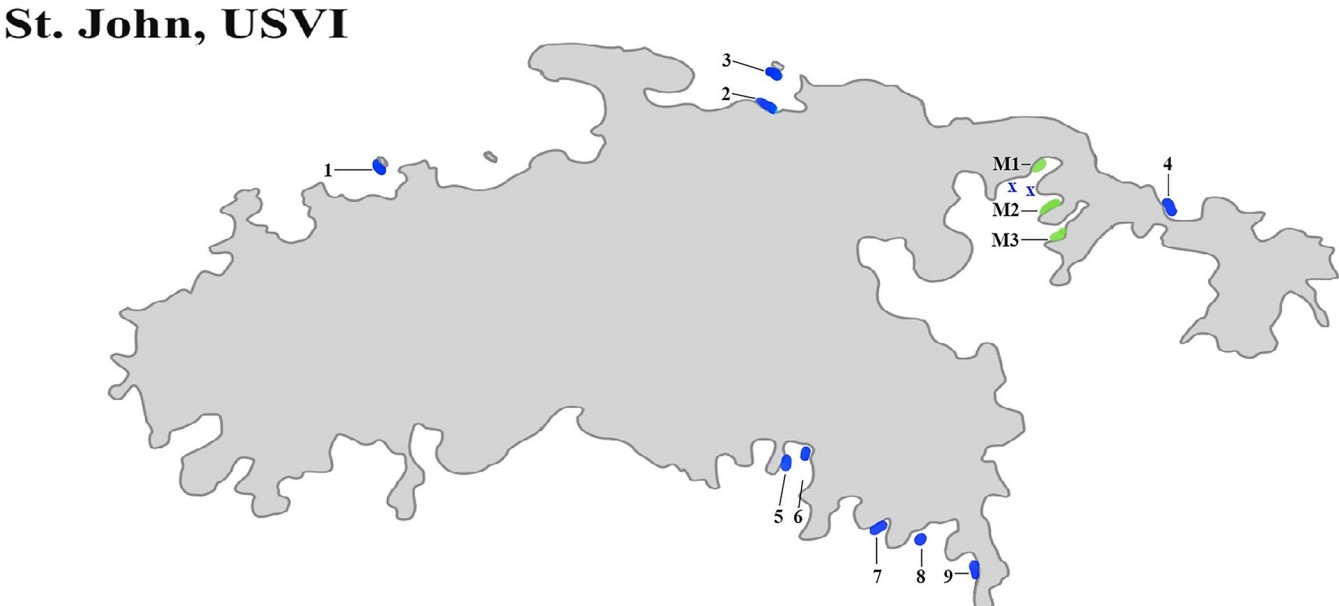

**Fig 2. Map STJ.** Survey sites in St John, USVI. Blue marks the numbered reef sites (1–9) and green marks the mangrove sites (M1-M3). The blue x's mark the two reef locations in Hurricane Hole that were anecdotally surveyed in 2022 and 2023 where urchin density remained high (2022 and 2023). Diadema Survey Map produced by Kristen Bell (Zenodo. https://doi.org/10.5281/zenodo.8372679).

2017 hurricanes, were considered among the most diverse mangrove sites in the Caribbean [39, 41].

Our surveys were conducted by snorkel, and like Carpenter and Edmunds [15], we targeted the "urchin zone" at each site. Specifically, before collecting data, we surveyed each site for 30 minutes, noting locations of *D. antillarum* aggregations. After the 30 min survey was completed, we counted all *D. antillarum* in three different 10 x 2 m transects. Each transect was separated by at least 15 m and selected for high aggregation. Because it is more difficult to safely place a weighted transect by snorkel in a way that avoids damaging benthic organisms, transects were established by a 10 m nylon rope stretched between two snorkelers above the substrate. All *D. antillarum* (juvenile and adult) within 1 m of either side of transect were then counted, including those hidden under rocks and crevices within the transect (see Fig 3A). The depth at each end of the transect was measured (to nearest 0.25 m) and the transect location was marked with a handheld global positioning system (GPS) unit (Garmin GPS 72H) in a waterproof bag. Mangrove sites were linear by nature, and the prop root fringe extended 1.5–2.5 m from the shoreline before the hurricanes. After the hurricanes, the mangrove fringe remained linear but at sites M2 and M3 the substrate was eroded and the prop root fringe was compressed. The transects included most of the urchin habitat within the prop roots given the distance of the fringe from the shoreline. Unlike the hard pavement that characterized the reef sites, the benthos of the mangrove fringe was primarily sand with scattered scleractinian coral heads.

After the 2022 disease outbreak, *D. antillarum* populations were extremely depleted at our sites (Fig 3B), prompting the establishment of transects whenever an urchin was encountered during the 30 min survey period. If more than three transects at a site were surveyed, the three transects with the highest counts were used for analysis. Water conditions (wave action and turbidity) and time limitations prevented population assessment at the following sites (year): 1 (2020); M1 (2019, 2020); M2 (2023); M3 (2023).

For each year at each reef site, we calculated the density (*D. antillarum* m$^{-2}$) by averaging the urchin counts from the three transects. As we were interested in comparing the impacts of the hurricanes and the disease outbreak on *D. antillarum*, we conducted two separate analyses

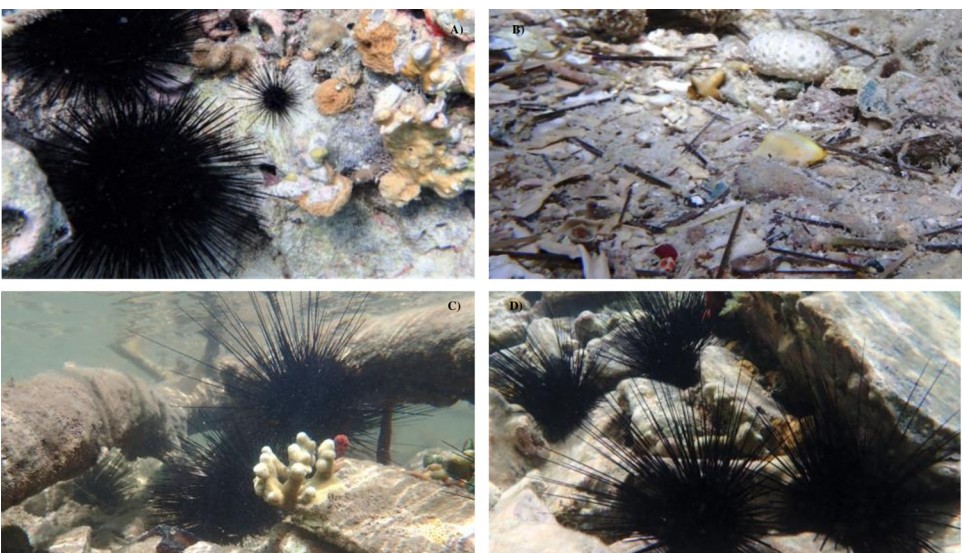

**Fig 3. Photos.** A) *D. antillarum* reef site 6 (2016, pre-hurricane) B) *D. antillarum* reef site 7, spines and test (2022) C) *D. antillarum* mangrove site M1 (2022) D) *D. antillarum* shallow reef adjacent to M1 (2022).

of the data from the reef sites. To assess the impact of the hurricanes on and subsequent recovery of *D. antillarum* populations, we compared the density in 2017 to that found in 2018 and 2019 using a repeated measure ANOVA with a Greenhouse-Geisser correction, as the data were normally distributed (Shapiro-Wilks p >0.05). Because we were unable to sample in 2021, we compared the data collected in 2020 to that collected in 2022 and 2023, a timeline that mimics that of the hurricanes (Fig 1). As these data were normally distributed, we used a repeated measures ANOVA with a Greenhouse-Geisser correction. All analyses were performed using IBM SPSS statistics Version 27 [42]. *D. antillarum* densities at the three mangrove sites in each sampling year were also calculated. The mangrove sample size prevented statistical comparisons, but patterns at reef sites were compared to those at mangrove sites graphically.

## Results

All our reef and mangrove transects were in shallow water (0.25–3.0 m), with most *D. antillarum* occurring at depths of 0.25–1.5 m and typically wedged into, or under, crevices generated by rubble, rocky pavement, and living scleractinian coral (Fig 3A, 3C and 3D). Occasionally, *D. antillarum* were found sheltering in the open in aggregations > 5 individuals, a pattern that was more likely to occur at mangrove sites and at reef sites before the hurricanes Irma and Maria. While we did not collect and measure *D. antillarum*, we did note variation in the size of adults across sites and years and consistently noted that the population at all sites was composed primarily of adults (>95%).

Prior to the hurricanes, *D. antillarum* density varied between reef sites (Fig 4). Some locations (sites 2, 4, 9) had fewer than 0.2 m$^{-2}$ while others ranged from 1.6 m$^{-2}$ (site 3) to 8.5 m$^{-2}$ (site 7, Fig 3A and 3C). After the hurricanes (2018), the average *D. antillarum* density dropped 66%, from the pre-hurricane (2017) average of 2.97 (S.E. ±0.98) m$^{-2}$ to 1.05 (±0.98) m$^{-2}$. Seven sites experienced moderate to dramatic declines in *D. antillarum* abundance, while two (sites 3 and 4) showed modest increases in abundance (Fig 5). By 2019, the density had increased at all sites, with two (sites 5 and 7) exhibiting relatively robust populations (> 3.5 m$^{-2}$). A repeated measures ANOVA revealed a marginally significant impact of time from hurricane on Diadema density (F (1.184,7) = 4.479, p = 0.057). Pairwise comparisons indicated a significant increase in *D. antillarum* density only between 2018 and 2019 (p = 0.042).

Populations in 2020 continued to increase at five sites (2, 3, 6, 7, 9; Fig 5), but five months after the disease event (2022), *D. antillarum* populations declined dramatically (96.4% reduction), from a site average of 2.51 (S.E. ±0.61) m$^{-2}$ to 0.09 (S.E. ±0.03) m$^{-2}$ (Fig 4). Though numerous *D. antillarum* tests and spines were found at each site (Fig 3B), no living urchins were found at site 2, density was less than 0.1 m$^{-2}$ at five locations (sites 1, 3, 5, 7, 9) and no locations had densities > 0.3 m$^{-2}$ (Fig 5). Eleven months later (2023), *D. antillarum* density had increased to 15% of the pre-disease average (0.39 ± 0.05 m$^{-2}$), with > 0.2 m$^{-2}$ found at all locations and > 0.4 m$^{-2}$ found at four sites (1, 2, 5, 8). A repeated measures ANOVA revealed a strongly significant impact of disease on urchin density (F (1.015,7) = 13.439, p = 0.006). Pairwise comparisons indicated significant differences in urchin density between all three sampling periods (2020:2022 p = 0.015; 2020:2023 p = 0.026; 2022:2023 p = 0.005).

*D. antillarum* populations in the mangroves appeared to exhibit a different pattern than those of the reef sites over the same sampling time period (Fig 6). Prior to the disease outbreak, mangrove sites had lower *D. antillarum* densities than reef sites; however, after the disease event (2022), one mangrove site (M1) had *D. antillarum* populations that were 7.5 times higher (2.1 ± 0.85 m$^{-2}$) than any reef site, and this high density persisted 16 months later (2023).

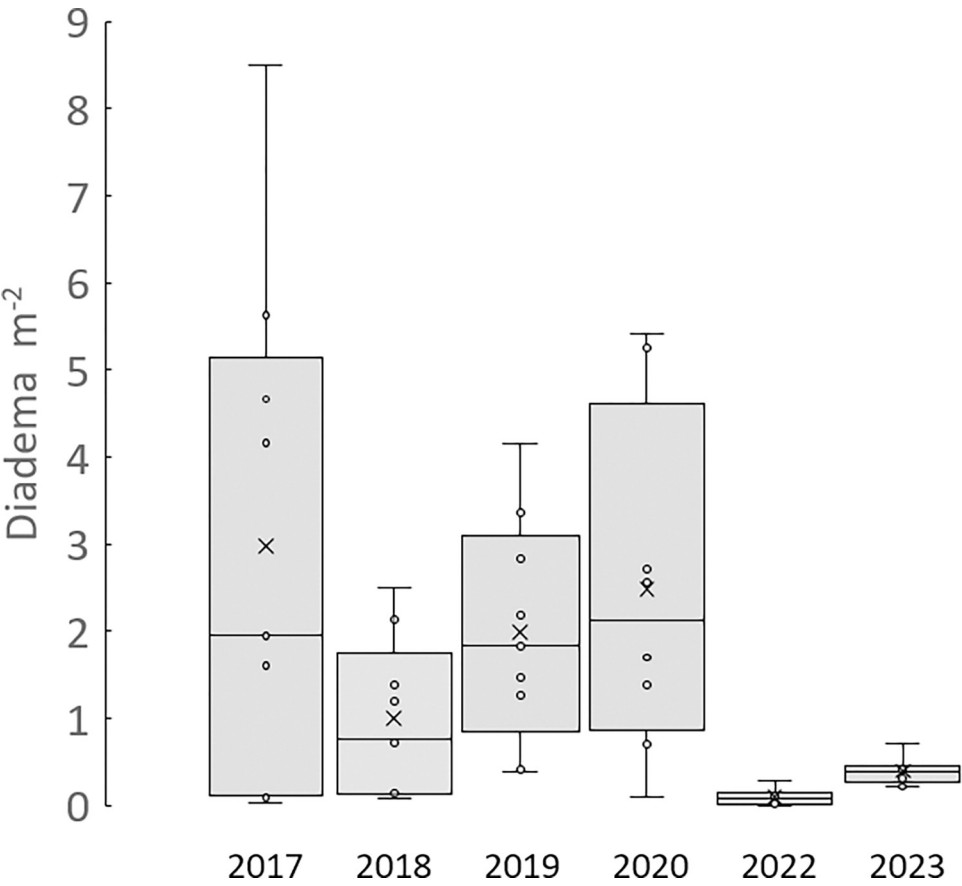

**Fig 4. Box-whisker overall reef sites.** Box-and-whisker plots of *D. antillarum* m⁻² at the nine reef sites in St. John sampled 2017–2023. The median value is indicated by the horizontal bar and the X marks the mean value.

## Discussion

Given that *D. antillarum* aggregate [43] and prefer to seek shelter in crevices [28] that can vary over small distances [44], we chose to survey *D. antillarum* in the aggregation areas which we determined by a standardized preliminary survey. As such, our density values represent a maximum abundance and don't lend themselves to comparison to values from studies using randomly placed transects. However, this method allows for reliable comparisons within sites and between years.

Prior to the back-to-back hurricanes of 2017, *D. antillarum* density at two thirds of our reef sites were similar to density patterns reported from the "urchin zones" sampled in Jamaica, St. Croix, Barbados, Belize, Bonaire, and Grenada [15]. Our overall site average of 2.97 (S.E. ±0.98) m⁻² indicated that, at least in pockets, *D. antillarum* were at densities that might support more successful reproduction in this prolific, yet asynchronous broadcast spawner. Five months after Hurricanes Irma and Maria, *D. antillarum* populations at the reef sites declined by 66%, like the losses reported after Hurricanes Hugo (1989) and Earl (2010) for *D. antillarum* at different depths (2–9 m) in Lameshur Bay [2]. And 17 months later, *D. antillarum* populations were increasing in density at all our sample sites in contrast to *D. antillarum* populations in the Florida Keys that exhibited a significant decline after Irma passed through at lower wind speeds (cat 4 in the Keys). Unlike populations in St. John, *D. antillarum* in the Florida Keys have remained at very low density and have shown very few signs of growth [45],

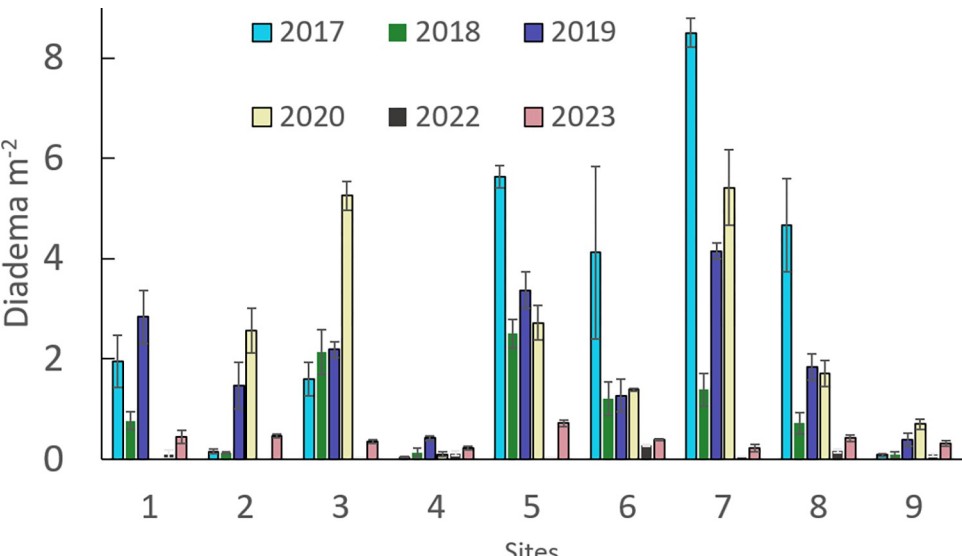

**Fig 5. Average individual reef sites.** Average (± S.E.) *D. antillarum* m$^{-2}$ (3 transects per site) for the 9 reef sites in St. John, USVI, sampled in 2017–2023. Wave action prevented sampling at Site 1 in 2020. Original data available (https://zenodo.org/records/8395238).

potentially leaving them more vulnerable to the impacts of hurricanes. A reduction of reef complexity because of a decrease in scleractinian coral health and abundance has been associated with reduced *D. antillarum* density [9, 46]. This reduction in reef crevices could reduce refugia for *D. antillarum*, making them more vulnerable to sediment abrasion, predation, and dislodgement [34, 47]. *D. antillarum* populations on reefs in southern Mexico remained robust after Hurricane Dean (2007), which could be attributed to the relatively high coral cover, which increases habitat complexity [28, 29]. The threat of hurricanes on struggling *D.*

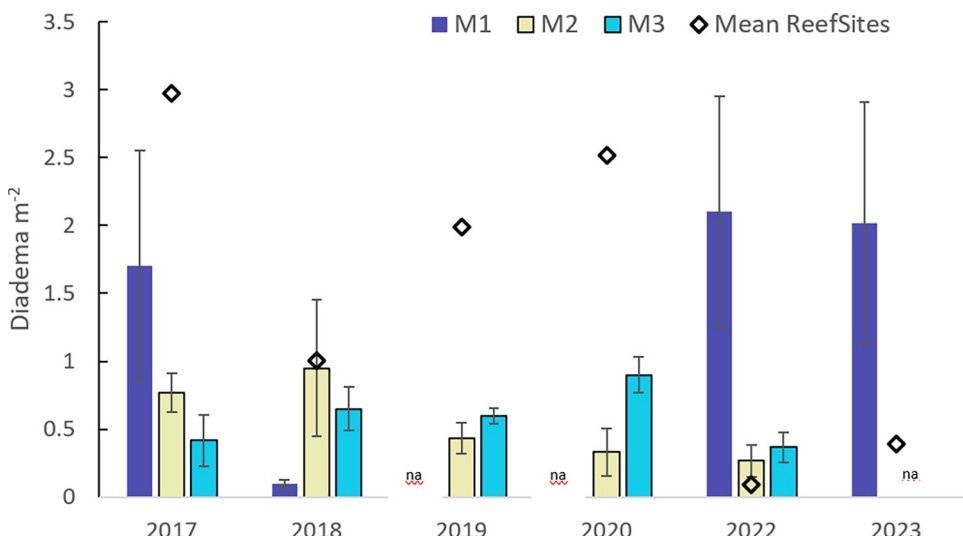

**Fig 6. Mangrove sites.** Average (± SE) *D. antillarum* m$^{-2}$ at the three mangrove sites (M1, M2, M3) 2017–2023. The diamond indicates the average *D. antillarum* m$^{-2}$ of the nine reef sites in each year. M1 was not sampled in 2019 or 2020 due to turbidity and debris, and in 2023 time constraints prevented sampling at M2 and M3. Original data available (https://zenodo.org/records/8395238).

PLOS ONE                                                                                                  Impact of hurricanes and disease on *Diadema antillarum*

*antillarum* populations remains of significant concern, as models predict that anthropogenic changes to the atmosphere are likely to increase the strength of hurricanes [48].

The 2022 disease outbreak devastated the rebounding *D. antillarum* population by 96.4% reduction, just slightly less than that reported for Saba, Caribbean Netherlands (99%) [35] and for deeper water sites in the Lameshur Bays, St. John, USVI (98%) [32]. These differences may be attributable to the fact that we targeted our samples to the "urchin zone" rather than along randomly placed transects, and all our sites were in shallow water (< 3 m). In 2023, we found modest but significant increases in *D. antillarum* density at all nine reef sites sixteen months after the disease outbreak was first noted. Overall, the *D. antillarum* population had increased to 0.39 m$^{-2}$, a mere 15% of the pre-disease density outbreak. While these density values remain low compared to pre-disease values, and remain at levels that challenge population growth, the increase in density at all reef sites was encouraging. We are aware of no other published studies that have reported patterns of *D. antillarum* density in 2023.

*D. antillarum* density at mangrove locations in Hurricane Hole appear to have followed a different pattern. Prior to the hurricanes, only one of the three mangrove sites (M1) had *D. antillarum* densities >1 m$^{-2}$. The western eye wall of Irma passed directly through Hurricane Hole and though the track of Maria was further south, this hurricane had a larger fetch area, which models suggest resulted in 2–3 m waves near the entrance of Hurricane Hole [38]. The combination of waves and winds from these two hurricanes uprooted mangrove trees, toppled coral, scoured the prop root communities, and transported rocks into shallow nearshore areas [39]. The wind and wave action also deposited human-made debris (e.g., boats, mooring lines, gasoline motors, etc.) in Hurricane Hole which further damaged many mangrove trees (authors' personal observations). Deposition of debris under the prop roots in the nearshore shallows of Otter and Water Creeks (M2 and M3, respectively) was more dramatic than at M1 and likely reduced habitat for organisms including *D. antillarum* [39]. Though damage was severe, *D. antillarum* density at M2 and M3 did not show dramatic declines post-hurricane, perhaps because pre-hurricane density was relatively low (< 1 m$^{-2}$). *D. antillarum* density at M1 exhibited a more dramatic decline (1.7 m$^{-2}$ to 0.1 m$^{-2}$). However in 2022, *D. antillarum* populations at M1 were remarkably high (> 2 m$^{-2}$) and populations at M2 and M3 were higher than at most reef sites. Unlike at reef sites, we did not observe clusters of *D. antillarum* tests and spines at the mangrove sites, but several dying *D. antillarum* (dropping spines) were noted beside an isolated coral head in the grassbed adjacent to M1, suggesting the disease was present in Hurricane Hole. Remarkably, *D. antillarum* density at M1 remained stable when resurveyed in 2023, suggesting that this population was somehow more resistant to the disease. In addition, we also noted robust populations of *D. antillarum* at two shallow water fringing reef sites located between grassbeds in Hurricane Hole (locations noted with an x in Fig 2). These two sites had notably less macroalgae than at any of our surveyed reef sites.

It is not clear why *D. antillarum* is faring so much better in Hurricane Hole than in other locations in St. John. Recent research has shown that extensive seagrass ecosystems can reduce the bacterial pathogen load in the water column and are associated with improved coral health [49]. Perhaps the extensive grassbeds in Hurricane Hole and Coral Bay (comprised primarily of *Thalassia testudinum*, *Syringodium filiforme*, and *Halophila stipulacea*) reduced the concentration of the scuticociliate that has been associated with the 2022 disease outbreak [36]. Scuticociliates are ubiquitous marine organisms and have not previously been associated with mass disease outbreaks in *D. antillarum*. Given that the disease first appeared near calm water ports and harbors, it is possible that these nutrient-rich environments may have fostered an explosive growth of a *Philaster*-like ciliate, which then dispersed rapidly throughout the Caribbean [36]. Much more research is needed to understand how ocean conditions, host factors, and

PLOS ONE | https://doi.org/10.1371/journal.pone.0297026   February 15, 2024                                                      8 / 12

other ecosystem actions might impact this pathogen and *D. antillarum* populations, both at local and Caribbean-wide levels.

The full extent of the impacts of the 2022 scuticociliate disease on *D. antillarum* populations remains to be seen. While some populations rebounded from the 1983 disease outbreak, the rebound was modest and remained an order of magnitude lower than pre-disease density in most locations. However, there has been an increase in restoration efforts Caribbean-wide, and artificial structures have been shown to support and maintain *D. antillarum* populations [12, 16]. This suggests that, if remaining *D. antillarum* find adequate refugia and nearest-neighbor density can be increased, it is possible that *D. antillarum* may rebound from this disease-outbreak. However, given the degraded state of Caribbean reefs and the increasing frequency of hurricanes, it is also possible that the remaining corals will continue to decline as algal-free zones disappear, further reducing the habitat viability for *D. antillarum* [50]. Only time will tell what the ultimate impact of this disease will be on *D. antillarum* populations.

## Acknowledgments

We thank Natasha Bestrom for help with field work and comments on a draft of the manuscript. We also thank Kristin Bell for producing the map of our study sites and Bonnie Bowers for advice on statistical analyses. We also thank Cheryl Taylor for technical support. Thanks to Thomas Kelley for help with obtaining research permits (VICR-2018-SCI-0001) to work in the Virgin Islands National Park and for the Virgin Islands Environmental Resource Station (VIERS) which provided support for research prior to the hurricanes in 2017.

## Author Contributions

**Conceptualization:** Renee D. Godard, C. Morgan Wilson, Natalie Badawy, Brittany Richardson.

**Formal analysis:** Renee D. Godard, C. Morgan Wilson.

**Funding acquisition:** Renee D. Godard.

**Investigation:** Renee D. Godard, C. Morgan Wilson, Caleb G. Amstutz, Natalie Badawy, Brittany Richardson.

**Methodology:** Renee D. Godard, C. Morgan Wilson, Caleb G. Amstutz, Natalie Badawy, Brittany Richardson.

**Supervision:** Renee D. Godard, C. Morgan Wilson.

**Visualization:** Renee D. Godard.

**Writing – original draft:** Renee D. Godard.

**Writing – review & editing:** Renee D. Godard, C. Morgan Wilson, Caleb G. Amstutz, Brittany Richardson.

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
