## [Decision Letter · Decision Letter 0]

5 Sep 2023

PONE-D-23-22614Impacts of hurricanes and disease on Diadema antillarum in shallow water reef and mangrove locations in St John, USVIPLOS ONE

Dear Dr. Godard,

Thank you for submitting your manuscript to PLOS ONE. After careful consideration, we feel that it has merit but does not fully meet PLOS ONE’s publication criteria as it currently stands. Therefore, we invite you to submit a revised version of the manuscript that addresses the points raised during the review process.

We look forward to receiving your revised manuscript.

Kind regards,

Jiang-Shiou Hwang, Ph.D.

Academic Editor

PLOS ONE

Journal Requirements:

   "We thank Natasha Bestrom for help with field work and comments on a draft of the manuscript. We also thank Kristin Bell for help with map and Bonnie Bowers for advice on statistical analyses. We also thank Cheryl Taylor for technical support. We are grateful to Tom and Anna Lawson for funding support for this project as well as the various internal funding support from Hollins University. Finally, thanks to Thomas Kelley for help with obtaining research permits (VICR-2018-SCI-0001) to work in the Virgin Islands National Park and for the Virgin Islands Environmental Resource Station (VIERS) which provided support for research prior to the hurricanes in 2017."

  "Funding for this research was provided by the Paula Pimlott Brownlee and Janet Spear Professorships at Hollins University as well as support from Tom and Anna Lawson.  The funders had no role in study design, data collection and analysis, decision to publish, or preparation of the manuscript."

4. We note that Figure 1 in your submission contain map/satellite images which may be copyrighted. All PLOS content is published under the Creative Commons Attribution License (CC BY 4.0), which means that the manuscript, images, and Supporting Information files will be freely available online, and any third party is permitted to access, download, copy, distribute, and use these materials in any way, even commercially, with proper attribution. For these reasons, we cannot publish previously copyrighted maps or satellite images created using proprietary data, such as Google software (Google Maps, Street View, and Earth). For more information, see our copyright guidelines: http://journals.plos.org/plosone/s/licenses-and-copyright.

Reviewers' comments:

Reviewer's Responses to Questions

**Comments to the Author**

1. Is the manuscript technically sound, and do the data support the conclusions?

Reviewer #1: Yes

Reviewer #2: Yes

2. Has the statistical analysis been performed appropriately and rigorously? 

Reviewer #1: Yes

Reviewer #2: Yes

3. Have the authors made all data underlying the findings in their manuscript fully available?

Reviewer #1: Yes

Reviewer #2: No

4. Is the manuscript presented in an intelligible fashion and written in standard English?

Reviewer #1: Yes

Reviewer #2: Yes

5. Review Comments to the Author

Reviewer #1: The authors aimed to explore the bioactivity response to changes in bacterial community in coral mucus and compare the difference of the bioactivities between the conditions of coral health and bleaching.

The topic is interesting and important, but the manuscript content quality is not good enough. In addition, the manuscript needs to be edited further as at many places the manuscript lack clarity.

About the survey locations, I recommend the author to provide exact latitude and longitude in the materials and methods, or use a tale to present this information. In the figure 1, the map is not informative. The reader cannot know the location of the St John, USVI.

About the sampling time, I suggest the authors to provide a time scale to illustrate the events such as observation time, hurricanes and the disease outbreak.

About the observation species Diadema antillarum, the author should provide a photo to show the morphology of it. Also, Authors need to present reasons for choosing this species in the beginning of the introduction.

I would suggest the authors to be more cautious to prepare the manuscript. Therefore, this article would need a thoughtful and concise structure, which this manuscript would need to revise before considering.

Reviewer #2: PONE-D-23-22614 – Impacts of hurricanes and disease on Diadema antillarum in shallow water reef and mangrove locations in St John, USVI

This paper presents the results of several years (2017-2023) of surveys at nine reef and three mangrove locations around St. John, USVI. Within these fixed sites, initial surveys identified any areas of Diadema abundance and targeted transects were used to generated a maximum density estimate. Overall, the manuscript appears technically sound and data support the conclusions. The information provided is of value. Clarity could be improved and/or more information provided in several areas. It is my opinion that the manuscript could be acceptable for publication following a major revision in which the authors address the below general and specific comments.

General comments:

- The Methods section makes clear (and the first paragraph of Discussion addresses) that data were collected from targeted searches designed to capture the greatest number of Diadema possible. This appears to be in line with previous similar studies and is valid in my opinion given the sparse and patchy nature of this species’ distribution. I wonder if the authors might consider including this facet of the methodology in the Abstract and/or Introduction? As a reader, if not made explicit, I generally assume that field data collection to “survey” or “document” was done in some type of random or haphazard fashion. I think simply including the term “targeted surveys” or something analogous would be helpful early on to set the expectation.

- It is unclear why the total surface area of each surveyed site was calculated (lines 158-161) or how this information was used. Lines 144-145 suggest that the densities reported in this study are based on the average number of urchins found within each of three 20 square meter transects per site (e.g. an average of 10 urchins per transect would result in a reported density of 0.5 urchins per square meter for that site). In my opinion, adding in the total site area calculation and results to the Methods may confuse the reader on which area was used as the denominator in urchin density calculations. Unless the total site area data are going to be elsewhere reported/analyzed beyond the Methods section, the relative total size of reef vs. mangrove sites could be mentioned only in the Discussion if it affects the way results were interpreted.

- More detailed description of mangrove site characteristics would be appreciated in the Methods and/or Results. How far out from the red mangrove fringe were transects taken? Was the benthos qualitatively similar to or different from the reef sites?

- In my opinion, there is a bit too much blending of anecdotal or qualitative observations with quantitative, statistically analyzed, comparisons. It could be made clearer when one vs. the other is being presented/discussed.

- Was any data on urchin size collected or is any anecdotal information available? This could add some inference as to population dynamics at these sites or differences between mangrove fringe and reef. Mangroves serve as nursery grounds for many reef-associated animals, so it’d be interesting to know if there were any differences in size between the habitat types.

- Data availability – the authors provide a DOI with the submission, but I do not see this linked within the article such that readers will have access following publication. Also, the data provided are means of three transects per site and not the full raw dataset.

Specific comments:

lines 42 – Should be “…disease outbreak on populations of…”?

line 55 – 71 per square meter would be at the extreme high end of observed densities. Suggest noting that this would have been an exceptional density or perhaps using the more “average” figures that are often reported.

lines 57 – Should be “that” not “which”.

line 67 – Females can likely produce 10 million or more eggs per spawn. See Table 1 in Pilnick et al. (2021) A novel system for intensive Diadema antillarum propagation as a step towards population enhancement. Eggs were collected from captive spawned animals, but it nonetheless reflects the fecundity potential of the species.

line 90 – Should be “that” not “which”.

lines 123-124 – Wording could be improved for clarity here. It seems like annual surveys were conducted in January from 2017 to 2020 and then in May/June in 2022 and 2023.

line 186 – Suggest making percentage directionally explicit within the parentheses – “(96.4% reduction)”. Also, suggest being consistent with significant digits on decimals – 0.092 should be 0.09.

line 213-214 – Is this from contemporary surveys of “urchin zones” in various places or from pre 1980s die off levels? Would help the reader to make this explicit in the text.

line 228 – Comma after “cover”.

lines 251-252 – This is the first time I am catching that surveys in mangrove areas may have been conducted right along the fringe within the prop roots of the trees themselves, or is this just suggesting that adjacent habitat that could have supported Diadema was lost? Do the authors think that other areas of mangrove fringe on St. John may serve as a “refugia” for D. antillarum, or is there something special about Hurricane Hole?

lines 262-263 – Do the blue x’s in Figure 1 represent anecdotal observations that were not included in data reported in this study? This should potentially be made explicit in the Figure 1 caption or elsewhere. Up until these lines, I had misunderstood those X’s as being the location of particularly high-density transects.

line 277 – I would argue that this study presents an impact of the 2022 die-off of D. antillarum populations. In this sentence, perhaps add “full extent of impacts” or something along those lines.

6. PLOS authors have the option to publish the peer review history of their article (what does this mean?). If published, this will include your full peer review and any attached files.

Reviewer #1: No

Reviewer #2: No

---

## [Author Response · Author response to Decision Letter 0]

14 Nov 2023

Responses to Academic Editor

1. Ensure that your manuscript meets PLOS ONE’s style requirements

We have followed the specific guidelines in the PLOS ONE style templates 

2. Remove Funding Information from the Acknowledgement section and ensure that the Funding Statement is accurate.

We have removed the funding information from the Acknowledgement section and confirm that the Funding Statement is accurate. 

We attempted to correct the mismatch in these two sections. The grants we received for this research were all internal institutional grants and were not available on the drop-down options of the site.

4. We note that Figure 1 in your submission contain map/satellite images which may be copyrighted

The map (now Figure 2) was developed by a colleague who is an artist. We have a signed form granting permission for use of the figure and the link to the uploaded image (https://zenodo.org/record/8372679)

5. The raw data was not fully available

We have uploaded the raw data which includes all transect counts for all years as well as lat-long for the sites. This original data is available (10.5281/zenodo.8395238) and we included this link in Fig 5 and 6.

Responses to Comments from REVIEWER #1

1. The summary statement from Reviewer 1, “The authors aimed to explore the bioactivity response to changes in bacterial community in coral mucus and compare the difference of the bioactivities between the conditions of coral health and bleaching,” does not apply to our work as we did not explore bacterial communities. 

2. Reviewer 1 was concerned about clarity, particularly in relationship to the timeline of the study. We appreciated their recommendation to provide a visual timeline. We developed a visual timeline which is now Fig 1 in our revised manuscript. In addition, we removed the potentially confusing language related to time in several locations in the manuscript (i.e. months after or before events) as this is now captured by the figure. 

3. Reviewer 1 suggested we include photos of the species to document morphology and evidence of disease. We have added Fig 3 (a,b,c,d) which shows: 3a clustered individuals before the hurricane; 3b remains (spines and tests) of urchins at a reef site in 2022; 3c cluster of urchins in mangrove 2022; 3d cluster of urchins at reef site adjacent to mangrove. 

4. Reviewer 1 suggested we remove the map and instead have Lat/Long. We feel that the map is useful but have included lat/long in the data available (10.5281/zenodo.8395238) 

Responses to Comments from REVIEWER #2

1. Reviewer 2 suggested we add language to the abstract that indicates we employed a targeted survey methodology

We did this on line 43 of abstract.

2. Reviewer 2 suggested we remove the surface area of the survey sites. 

We did this for clarity. 

3. Reviewer 2 requested that we add more detail on the mangrove site characteristics, including the benthos.

We did this on lines 154-159 of marked up manuscript.

4. Reviewer 2 suggested we be clearer in the results

We clarified the results in several locations including removing one anecdotal observation.

5. Reviewer 2 asked for additional information on adult to juvenile ratio and potential differences between reef and mangrove sites. We added clarifying language and have included that language 206-208 (marked manuscript) indicating the vast majority of the D. antillarum were adults and no difference was noted in the abundance of juveniles between coral and mangrove sites. 

6. Reviewer 2 had a number of specific editorial comments. Our responses to each are noted in blue

lines 42 – Should be “…disease outbreak on populations of…”? Corrected

line 55 – 71 per square meter would be at the extreme high end of observed densities. Suggest noting that this would have been an exceptional density or perhaps using the more “average” figures that are often reported. Included the range originally reported 3 up to 71 (now lines 56-57 of marked manuscript)

lines 57 – Should be “that” not “which”. Corrected

line 67 – Females can likely produce 10 million or more eggs per spawn. See Table 1 in Pilnick et al. (2021) A novel system for intensive Diadema antillarum propagation as a step towards population enhancement. Eggs were collected from captive spawned animals, but it nonetheless reflects the fecundity potential of the species. Interesting – we changed language to “can produce millions of eggs per spawn” (line 69 of marked manuscript)

line 90 – Should be “that” not “which”. Corrected

lines 123-124 – Wording could be improved for clarity here. It seems like annual surveys were conducted in January from 2017 to 2020 and then in May/June in 2022 and 2023. Timeline clarified this and language was improved

line 186 – Suggest making percentage directionally explicit within the parentheses – “(96.4% reduction)”. Also, suggest being consistent with significant digits on decimals – 0.092 should be 0.09. Corrected

line 213-214 – Is this from contemporary surveys of “urchin zones” in various places or from pre 1980s die off levels? Would help the reader to make this explicit in the text. clarified

line 228 – Comma after “cover”. Corrected

lines 251-252 – This is the first time I am catching that surveys in mangrove areas may have been conducted right along the fringe within the prop roots of the trees themselves, or is this just suggesting that adjacent habitat that could have supported Diadema was lost? Do the authors think that other areas of mangrove fringe on St. John may serve as a “refugia” for D. antillarum, or is there something special about Hurricane Hole? Clarified the description of mangrove locations in the methods (now lines 154-159 in marked manuscript). There are no other robust mangrove locations outside of Hurricane Hole. Perhaps Hurricane Hole is unique in serving as a refugia certainly other mangrove locations in the Caribbean should be surveyed.

lines 262-263 – Do the blue x’s in Figure 1 represent anecdotal observations that were not included in data reported in this study? This should potentially be made explicit in the Figure 1 caption or elsewhere. Up until these lines, I had misunderstood those X’s as being the location of particularly high-density transects. Clarified the explanation in the text as well as in the figure legend (now lines 142-143 of marked manuscript).

line 277 – I would argue that this study presents an impact of the 2022 die-off of D. antillarum populations. In this sentence, perhaps add “full extent of impacts” or something along those lines. Yes – thanks for the suggestion – corrected (now line 542 of marked manuscript).

---

## [Decision Letter · Decision Letter 1]

27 Dec 2023

Impacts of hurricanes and disease on Diadema antillarum in shallow water reef and mangrove locations in St John, USVI

PONE-D-23-22614R1

Dear Dr. Godard,

We’re pleased to inform you that your manuscript has been judged scientifically suitable for publication and will be formally accepted for publication once it meets all outstanding technical requirements.

Kind regards,

Jiang-Shiou Hwang, Ph.D.

Academic Editor

PLOS ONE

Additional Editor Comments (optional):

Reviewers' comments:

Reviewer's Responses to Questions

**Comments to the Author**

1. If the authors have adequately addressed your comments raised in a previous round of review and you feel that this manuscript is now acceptable for publication, you may indicate that here to bypass the “Comments to the Author” section, enter your conflict of interest statement in the “Confidential to Editor” section, and submit your "Accept" recommendation.

Reviewer #1: All comments have been addressed

Reviewer #2: All comments have been addressed

2. Is the manuscript technically sound, and do the data support the conclusions?

Reviewer #1: Yes

Reviewer #2: Yes

3. Has the statistical analysis been performed appropriately and rigorously? 

Reviewer #1: N/A

Reviewer #2: Yes

4. Have the authors made all data underlying the findings in their manuscript fully available?

Reviewer #1: Yes

Reviewer #2: Yes

5. Is the manuscript presented in an intelligible fashion and written in standard English?

Reviewer #1: Yes

Reviewer #2: Yes

6. Review Comments to the Author

Reviewer #1: In this revision, since the author provided enough data/explanation to all comments, the manuscript is acceptable. However, I would suggest the author to provide figures with higher resolution before publication.

Reviewer #2: Thank you for thoroughly addressing reviewer comments. It is my opinion that the article is acceptable for publication.

7. PLOS authors have the option to publish the peer review history of their article (what does this mean?). If published, this will include your full peer review and any attached files.

Reviewer #1: No

Reviewer #2: **Yes: **Joshua Patterson

---

## [Editor Report · Acceptance letter]

6 Feb 2024

PONE-D-23-22614R1 

PLOS ONE

Dear Dr. Godard, 

I'm pleased to inform you that your manuscript has been deemed suitable for publication in PLOS ONE. Congratulations! Your manuscript is now being handed over to our production team.

Kind regards, 

on behalf of

Prof. Jiang-Shiou Hwang 

Academic Editor

PLOS ONE